# Transcriptome Dynamics Underlying Magnesium Deficiency Stress in Three Founding *Saccharum* Species

**DOI:** 10.3390/ijms23179681

**Published:** 2022-08-26

**Authors:** Yongjun Wang, Yihan Li, Xiuting Hua, Zhe Zhang, Tianqu Fan, Wei Yao, Muqing Zhang, Jisen Zhang

**Affiliations:** 1Center for Genomics and Biotechnology, Fujian Provincial Key Laboratory of Haixia Applied Plant Systems Biology, Key Laboratory of Sugarcane Biology and Genetic Breeding, National Engineering Research Center for Sugarcane, Fujian Agriculture and Forestry University, Fuzhou 350002, China; 2Guangxi Key Laboratory of Sugarcane Biology, Guangxi University, Nanning 530004, China

**Keywords:** sugarcane, transcriptome dynamics, magnesium deficiency, WGCNA, response mechanism

## Abstract

Modern sugarcane cultivars were generated through interspecific crossing of the stress resistance *Saccharum spontaneum* and the high sugar content *Saccharum officinarum* which was domesticated from *Saccharum robustum*. Magnesium deficiency (MGD) is particularly prominent in tropical and subtropical regions where sugarcane is grown, but the response mechanism to MGD in sugarcane remains unknown. Physiological and transcriptomic analysis of the three founding *Saccharum* species under different magnesium (Mg) levels was performed. Our result showed that MGD decreased chlorophyll content and photosynthetic efficiency of three *Saccharum* species but led to increased starch in leaves and lignin content in roots of *Saccharum *robustum** and *Saccharum *spontaneum**. We identified 12,129, 11,306 and 12,178 differentially expressed genes (DEGs) of *Saccharum *officinarum**, *Saccharum *robustum** and *Saccharum *spontaneum**, respectively. In *Saccharum *officinarum**, MGD affected signal transduction by up-regulating the expression of xylan biosynthesis process-related genes. *Saccharum *robustum,** responded to the MGD by regulating the expression of transcription and detoxification process-related genes. *Saccharum *spontaneum**, avoids damage from MGD by regulating the expression of the signing transduction process and the transformation from growth and development to reproductive development. This novel repertoire of candidate genes related to MGD response in sugarcane will be helpful for engineering MGD tolerant varieties.

## 1. Introduction

Modern cultivars sugarcane (*Saccharum* spp.) is a widely grown crop with high biomass. It possesses high photosynthetic efficiency, low CO_2_ compensation points and abiotic stress resistance [1]. Modern cultivars sugarcane is cultivated on ~26 million hectares of land in more than 90 countries all over the world with 1.84 billion metric tons harvested annually [2]. It contributes approximately 80% of sugar to the sugar industry and generates approximately 40% of ethanol produced worldwide [3], and therefore attracts much attention as an alternative biofuel source [4]. Modern sugarcane cultivars are founded on interspecific hybrids between *S. spontaneum* (2*n* = 40–128) and *S. officinarum* (2*n* = 80)*,* which were then subjected to repeated backcrosses to *S. officinarum* [5]. In addition, *S. officinarum* is thought to be domesticated from *S. robustum* (2*n* = 60–200) [6,7].

Sugarcane is mainly cultivated in tropical and subtropical regions (31° south to 36.7° north) [8] and approximately 1.7 billion hectares (nearly 1/3) have acidic soils in these regions [9,10]. The magnesium ion (Mg^2+^) is taken up by plants and due to it having the smallest ionic radius but the largest hydrated radius among cations, Mg^2+^ binds weakly in soil and root cell walls in low pH environment, which leads to the loss of the Mg easily from soil rendering it readily exchangeable [11]. Furthermore, the uptake rate of Mg^2+^ is also strongly depressed by competition from cations, such as NH^4+^ [12], Ca^2+^ [13] and Mn^2+^ [14] in acid soil [15]. The above reasons lead to Mg deficiency (MGD) which is particularly prominent in tropical and subtropical areas.

An Mg level of about 0.15–0.35% of the dry weight is required to participate in numerous physiological and biochemical reactions for plant growth and development [16]. In plants, Mg plays an essential role in plant physiological and biochemical processes, including energy balance, signaling and protein synthesis, and it can significantly affect the yield and quality of crops [17]. As the central element of chlorophyll, Mg is a co-factor required to activate more than 300 enzymes [11]. Even though Mg plays an essential role in plant development and growth, it is given little attention compared to other macronutrients. As a result, Mg is also called “the forgotten element” [10].

MGD in plants is a widespread problem affecting productivity and quality in agricultural systems and forestry. Several studies have investigated and uncovered the effects of MGD on plant physiology, including biomass partitioning, CO_2_ assimilation, photo-oxidative defense, net CO_2_ assimilation and biomass. MGD triggers nutrient translocation and the expression of genes involved in the protection response in young leaves [18]. MGD reduced the yield and seed germination in the wax gourd by affecting the carbohydrate translocation [19]. Statistical data shows that MGD led to a decreased biomass of 39% compared to Mg supplemented (MGS) treatment plants, and the concentration of reactive oxygen species (ROS) was increased by about 69% in MGD compared to MGS [20]. When grown under the optimum conditions of potassium, phosphorus and magnesium, the sugarcane yield will increase significantly, demonstrating Mg plays critical role in the growth and development of sugarcane [21]. However, there is no report of MGD being part of the molecular mechanism. With the recent availability of the sugarcane genome [2], global transcriptome analysis of MGD in sugarcane has become a cost-effective way to study the molecular mechanism in plants. In this study, to explore the molecular basis of the sugarcane response to MGD, we investigated the phenotypic and physiological data of sink and source tissues in the three *Saccharum* species treated with MGD and analyzed the transcriptome dynamics and the relative co-expressed gene network based on the RNA-seq of the leaf and root with different treatments. The results provide a valuable foundation for the genetic improvement of sugarcane under MGD.

## 2. Results

### 2.1. Phenotypic Observation of Three Saccharum Species under Different Mg^2+^ Treatment

Phenotypic analysis result show that three *Saccharum* species (*S. officinarum*, *S. robustum* and *S. spontaneum*) displayed leaf chlorosis and decreased amounts of roots under 0 mM Mg^2+^ treatment (MGD) (Figure 1). In leaves under MGD, net photosynthesis efficiency (PN) and SPAD values were significantly decreased in the three *Saccharum* species (Figure 1C,D). Starch content in leaves under MGD of *S. spontaneum* was significantly higher than 0.2 mM Mg^2+^ treatment (control condition) and 1 mM Mg^2+^ treatment (HMT) (Figure 1E) but had no difference in leaves of *S. officinarum* (Figure 1E), respectively. To further study the effects of MGD on the roots of sugarcane, *the relative* lignin contents of three *Saccharum* species treated with three Mg concentration were determined. Compared with control condition and HMT plants (≤9.41), relative lignin contents in roots of three *Saccharum* species under MGD were significantly increased (10.32–11.90), while relative lignin content in roots at control condition had no obvious difference when compared to HMT (Figure 1F).

### 2.2. Transcriptome Sequencing and Gene Expression Analysis

A total of 51 RNA-seq libraries of the leaves and roots from the three *Saccharum* species were used for RNA -seq sequencing. In total, 1126.77 million, 150 bp paired-end reads (~220.72 Gb) were aligned to the *S. spontaneum* reference genome by Cufflinks [2] (Appendix A). The mapping rates ranged from 75.99% to 95.73% (Appendix A). Hierarchical clustering of samples based on FPKM showed the correlation coefficients ranged from 0.6 to 1.0 in the same tissues with the same condition (Appendix A). Principle components analysis (PCA) of the gene expressional patterns were consistent with the genetic separation among three *Saccharum* species (Appendix A).

### 2.3. The Divergence of Transcriptome among the Three Species under Treatment of Mg^2+^

Under the three levels of Mg^2+^ treatments, we identified 12,129, 11,306 and 12,178 DEGs in *S. officinarum,*  *S. robustum* and *S. spontaneum*, respectively (Figure 2, Appendix A). There are 8836/7839/11,920 DEGs in leaves and 8703/7431/3983 DEGs in root of *S. officinarum,* *S. robustum* and S. spontaneum, respectively (Figure 2A–C). Of these DEGs, 4973 DEGs were present in all *Saccharum* species (Figure 2D) and 2960, 2037 and 3037 were present specifically in *S. officinarum*, *S. robustum* and S. spontaneum, respectively (Figure 2D).

### 2.4. GO Category Enrichment of DEGs among Three Saccharum Species under MGD Stress

To analyze the function of the DEGs among three *Saccharum* species, we performed the GO enrichment analysis using Fisher’s exact test with *p* value ≤ 0.001 as the cutoff. In accordance with the K-Mean clustering algorithm, we clustered all DEGs in the leaves and roots of the three *Saccharum* species into six main clusters (designated Cluster 1–6) depending on their expression in different magnesium concentrations (Figure 3A,C, Appendix A). Three groups of genes with similar trends were detected in all samples (Figure 3A,C): (1) a pattern of up-regulation under MGD and control condition was detected in C1 and C2 of all *Saccharum* species and C3 in leaves of *S. robustum*; (2) while C4 and C5 in leaves of *S. robustum* and the C3 and C4 of other samples exhibited down-regulated gene expression profiles; (3) DEGs in C6 of all samples and C5 in leaves of *S. spontaneum* were up-regulated under control condition and down-regulated in MGD and HMT. In roots the DEGs displayed the opposite expression trends in C5 and C6.

GO enrichment analysis showed that DEGs with similar trends were enriched in the same categories (Figure 3B,D). The response to stimulate related DEGs is clearly up-regulated under MGD in the leaves and roots of three *Saccharum* species. In contrast, the DEGs related to the chloroplast in leaves (e.g., Chloroplast thylakoid membrane and Chloroplast envelope) and membrane (e.g., Plasma membrane, Cell periphery and Anchored component of membrane) in roots were down-regulated under MGD in all three *Saccharum* species.

In leaves (Figure 3B), DEGs have similar functional categories in *S.*  *robustum* and *S. officinarum*. Compared to the *S. *robustum** and *S. officinarum*, the apoplast, cytosol, plant-type vacuole and ATPase activity processes are specific to *S. *spontaneum** up-regulated DEGs only and are absent in the other two *Saccharum* species. Genes encoding DNA metabolic processes and DNA binding are significantly enriched in down-regulated groups of *S. spontaneum*. In miscellaneous groups, we found that the stress response in *S. officinarum*, chloroplast thylakoid membrane in *S. robustum* and photosynthesis, chloroplast and thylakoid in *S. spontaneum* were up-regulated in control condition, and down-regulated in MGD and HMT.

In roots (Figure 3D), the purine metabolic process was specifically enriched by the up-regulated DEGs of *S. officinarum*. Ion transport processes (for example transition metal ion homeostasis, ion transmembrane transport) and transcription process related genes were present in the up-regulate DEGs under MGD of *S. robustum*. The down-regulated group of DEGs of *S. officinarum* were present in the extracellular region and plant-type cell wall clusters. The inorganic ion transmembrane transporter, monovalent inorganic cation transport, cell periphery and transmembrane signaling receptor activity clusters were only identified as down-regulated DEGs in *S. robustum*. In *S. spontaneum*, the DEGs were mainly enriched in the oxidation-reduction process and circadian rhythm clusters. In the miscellaneous group, the stimulation response process in *S. officinarum*, cytosolic large ribosomal subunit in *S. *robustum**, transcription process (for example nucleolus and ribosomal subunit) and mitochondrial inner membrane in *S. spontaneum* were down-regulated in control condition. While the stress response process and transmembrane signaling receptor activity in *S. robustum*, and membrane in *S. spontaneum* were up-regulated under the control condition.

Furthermore, we performed the comparative analysis of GO category enrichment among the three *Saccharum* species (Figure 3E). Our data showed that the DEGs of three *Saccharum* species had shared functional clusters including chloroplast envelope, chloroplast stroma, oxidation-reduction process, response to water deprivation and the malate metabolic process. Consistent with the above evidence, we found that functional categories present in each *Saccharum* species also have a similar distribution (Figure 3B,D,E). For example, the stress response in *S. officinarum* (e.g., response to wounding), ion transport process in *S. robustum* (e.g., ion transmembrane transport) and Photosynthetic in *S. spontaneum* (e.g., photosynthetic electron transport chain). These results also add support to a response to MGD that is mechanistically shared of the three *Saccharum* species.

### 2.5. Mapman Annotation DEGs under MGD

A representative monoploid gene of *S. spontaneum* was annotated by Mapman and 22,720 genes (62.43%) were annotated in 34 pathways (Appendix A). The DEGs in the three *Saccharum* species were mainly involved in five functional categories, including protein (average 16.29%), RNA (average 14.94%), miscellanous (average 13.92%), transport (average 9.68%) and stress (average 6.71%) (Appendix A).

We compared DEGs top 5 enriched functions under three Mg concentration treatment between leaves and roots of the three *Saccharum* species (Figure 4A, Appendix A). In leaves, 6.99–16.44% of the up-regulated genes of the three *Saccharum* species under MGD were involved in protein, RNA, miscellaneous and transport. The secondary metabolism DEGs were enriched in leaves of *S.*  *officinarum* (7.56%) and *S. robustum* (8.33%). Stress (8.6%) and lipid metabolism (8.06%) were enriched in the DEGs of *S. spontaneum* (Appendix A). Of the down-regulated DEGs under MGD, in *S.* *officinarum*, hormone metabolism and signaling pathways, were enriched by 8.41% and 7.48%, respectively. Stress (7.55%) and lipid metabolism (7.55%) were the main clusters enriched in *S. robustum* and stress (11.54%) and hormone metabolism (9.62%) were the main clusters enriched in *S. spontaneum*.

In roots, 5.56–27.78% of up-regulated genes under MGD were involved in protein, RNA, miscellaneous, transport, stress and hormone metabolism (Figure 4A, Appendix A). 6.45% and 5.56% of DEGs were involved in signaling in *S. robustum* and *S. spontaneum*, respectively. Secondary metabolism (5.56%) and tetrapyrrole synthesis (5.56%) were mainly enriched in *S. spontaneum*. In summary, among three *Saccharum* species, the DEGs in leaves and roots under MGD have the same enrichment patterns in RNA and protein processes while displaying significant differences in the stress response, signal transduction and secondary metabolism (Appendix A).

We found that the following genes were up-regulated under MGD (Appendix A): potassium transporter (*HAK*, *High-affinity K+ transporter*, *Sspon.001A0027724*), starch synthase (*SS2*, *Starch synthase 2*, *Sspon.004A0004511*), circadian rhythm (*LHY*, *LATE ELONGATED HYPOCOTYL*, *Sspon.006A0014990*), senescence and high-light acclimation (*FTSH6*, *FTSH PROTEASE 6*, *Sspon.008A0014242*) and stress response (*FBA2*, *FRUCTOSE-BISPHOSPHATE ALDOLASE 2*, *Sspon.005A0023720*). However, the stress response (*COR27*, *COLD REGULATED GENE 27*, *Sspon.001A0005031*), thioredoxin localized in chloroplast stroma (*WCRKC1*, *WCRKC thioredoxin 1*, *Sspon.004A0003040*), response to the level of sugar in the cell (*DIN10*, *DARK INDUCIBLE 10*, *Sspon.008B0017730*) and circadian clock (*PCL1*, *PHYTOCLOCK 1*, *Sspon.003B0000252*) were down-regulated by MGD. The transcription factors (TFs), such as MYB (*Sspon.006D0013430*), DOF (DNA-binding with one finger), zinc finger (*Sspon.003A0006040*) and WRKY (*Sspon.002A0018720*) family were suppressed under MGD conditions (Appendix A).

We further identified the DEGs that participate in starch biosynthesis and metabolic pathways to enable exploration of the molecular mechanisms of variation of starch accumulation under MGD in leaves of the three *Saccharum* species (Figure 4B). We observed that in *S. spontaneum,* the up-regulated genes under MGD were involved in glucose-1-phosphate adenylyl transferases (*Sspon.006A0010082*, *Sspon.007A0001672* and *Sspon.007C0000870*), granule-bound starch synthase (*Sspon.002A0025930*), starch branching enzyme (*Sspon.008A0001272*)*,* these genes were slightly induced in *S. officinarum* but were not expressed in *S. robustum*.

### 2.6. Co-Expression Networks Analysis by WGCNA

In leaves, 9182 DEGs were used for WGCNA analysis, and 16 co-expression modules were identified (Appendix A). The Mg^2+^ concentration positive module (lightcyan1, darkorange and black) in leaves had a highly positive with SPAD value, net photosynthesis rate and transpiration efficiency (Figure 5A, Appendix A), while the lightcyan1 module (R = −0.4, *p*−value = 0.04) and orangered4 module (R = −0.44, *p* value = 0.02) had negative correlation with the starch content in leaves. The Mg^2+^ concentration negative modules lavenderblush3 module (R = −0.61, *p* value = 7 × 10^−4^) and violet module (R = −0.44, *p* value = 0.02) were negatively correlated with the SPAD value, net photosynthesis rate and transpiration efficiency. According to the module−species correlation analysis, we established that the lightcyan1 and lavenderblush3 module responded to different Mg^2+^ concentration treatments in the three *Saccharum* species. The orangered4 module (R = 0.57, *p*−value = 0.002) and violet module (R = 0.68, *p*−value = 8 × 10^−5^) correlated highly with *S. officinarum*, the darkorange module (R = 0.47, *p*−value = 0.01) and coral2 module (R = 0.39, *p*−value = 0.04) were positively correlated with *S. robustum* and the black module (R = 0.78, *p*−value = 2 × 10^−6^) were positively correlated with *S. spontaneum*. GO analysis showed that the function of the “stress response”, “carboxylic acid transmembrane transport” and “metabolism process” were enriched among three *Saccharum* species under different Mg^2+^ concentration treatments (Table 1). In *S. officinarum*, the Mg^2+^ concentration association module was enriched in the photosynthesis pathway. In *S. robustum*, the Mg^2+^ concentration associate module was enriched in “response to organic cyclic compound”, “circadian rhythm”, “response to abscisic acid,” and “ribosomal large subunit biogenesis”, “cytoplasmic translation” and “dicarboxylic acid metabolic process”. The specific DEGs of *S. spontaneum* were enriched in “response to hypoxia”, “rRNA 5′-end processing,” and “photo morphogenesis” (Table 1). The above results indicate that Mg^2+^ plays an important role in photosynthesis, circadian rhythm, metabolism and chloroplast development in sugarcane leaves and there are a series of feedback steps to respond to the MGD/HMT and maintain Mg^2+^ homeostasis in sugarcane leaves.

In roots, 6691 DEGs were used for WGCNA analysis, and 14 co-expression modules were identified (Appendix A). Two modules named lightgreen (R = −0.39, *p* value = 0.04) and magenta (R = −0.39, *p* value = 0.05) were negatively associated with Mg^2+^ concentration and positivity with the lignin content (Figure 5B). Species−module associated results show that lightgreen modules have a highly positive association with *S. officinarum* (R = 0.68, *p* value = 1 × 10^−4^), and are negatively correlated with *S. spontaneum* (R = −0.47, *p* value = 0.01). The magenta module correlated highly positively with *S. robustum (*Figure 5B). GO enrichment analysis showed that the lignin biosynthesis process (GO:0009808, GO:0045492 and GO:0009699) was shared between lightgreen and magenta modules (Table 2). Moreover, the small molecule catabolic process was enriched in the lightgreen module and the DEGs of the magenta module were enriched in inorganic substances and amine metabolic processes (Table 2). In roots, these data indicated that the Mg^2+^ concentration-response genes are different among three *Saccharum* species with the response genes of *S. officinarum* being clustered in the lightgreen module and the *S. robustum* is magenta.

### 2.7. Hub Genes Investigating by WGCNA

In leaves, the DEGs involved in “transporter”, “photosynthesis”, “chloroplast development”, “circadian rhythm, synthesis”, “metabolism of carbohydrate” and “transcription factors” were used to construct the gene co-expression network (Figure 5C–I). In these networks, 117 transcription factors (TFs) have co-expression with downstream genes. Based on the degree of connectivity, the top 1/10 genes of each module were regarded as hub genes, a total of 43 hub genes were identified in the Mg^2+^ concentration positive associated modules and 12 TFs were identified as the hub genes (Figure 5C–F, Appendix A), which belong to TCP (KNAT3, TCP8, TCP14), NAC (NAC028, NAC041), GATA (GATA5), ERF (ERF060), CO-like (BBX12, COL9), G2-like (PCL1, HHO3) and TALE (BLH2) families. Additionally, other hub genes were involved in chloroplast development (18/43), photosynthesis (7/43), rhythm (4/43), synthesis, metabolism of carbohydrate and transport. 18 hub genes were identified in Mg^2+^ concentration negative associated modules and 59 TF had co-expression for downstream genes and 8 of these were identified as hub genes (Figure 5G–I, Appendix A), which belong to WRKY (WRKY18, WRKY33), NAC (ANAC087), C3H (ZFN3), HD-ZIP (HB9), bZIP (bZIP19), Dof (CDF2) and MYB(MYB5) families. The statistical analysis of hub genes according to the functional classification, revealed that hub genes were involved in photosynthesis (12/61), circadian rhythm (6/60), synthesis and metabolism of carbohydrate (2/60) (Figure 5C–I, Appendix A) with the exception of TFs (20/61).

In roots, we used the genes involved in “response to stimulus”, “transcription factors”, “transporter” and “lignin biosynthesis and metabolic process” to construct a gene correlation network and we found that the hub genes of these modules were associated with transport (1/8), response to stimulus (6/8) and lignin biosynthesis (1/8) in lightgreen module. In the magenta module, 6 TFs were identified as hub genes including BHLH51 (*Sspon.003D0026630*), NAC041 (*Sspon.006A0001661*), HB-7 (*Sspon.004A0009103*), MYB (*Sspon.003B0032860*) and NF-YA6 (*Sspon.001B0034221*), NF-YA10 (*Sspon.002B0038780*), other hub genes were distributed in response to stimulus (4/23), and transporter (5/23) categories (Figure 5B, Table 2, Appendix A).

### 2.8. Verification of Gene Expression Based on RNA-Seq by qRT-PCR

Three DEGs including an Mg^2+^/H^+^ exchanger (*MHX1*, *Sspon.006D0023280*), High-affinity K^+^ transport 1 (*HKT1*, *Sspon.008A0002872*) and Magnesium transport 10 (*MGT10*, *Sspon.003A0000870*) were selected for quantitative real-time PCR (qRT-PCR) verification. The result showed that the expression patterns detected with qRT-PCR were consistent with the RNA-seq analysis (R^2^ = 0.9195), indicating the reliability of RNA-seq expressional profiles (Appendix A).

## 3. Discussion

This study compared the physiological and biochemical indexes and transcriptomes under three different Mg^2+^ concentration treatments among three *Saccharum* species. Based on the physiological and biochemical analysis, we found that MGD significantly reduced the chlorophyll content in sugarcane leaves and caused a sharp decrease in photosynthetic efficiency (Figure 1C,D). The phenotypic observation indicated that MGD led to leaf chlorosis, shortened stalks and reduced root mass, although the root hair density increased in sugarcane (Figure 1A,B). These phenomena confirmed the significant effect of Mg^2+^ treatment for the *Saccharum* species.

Starch metabolism plays a crucial role in leaves of C_4_ plants to avoid triphosphate limitation, as it accelerates the exportation of photosynthetic products compared to sucrose synthesis export [22]. Mg plays a crucial role in phloem loading and export of photosynthates, including sucrose from leaves. Leaf carbohydrates, including starch, accumulate due to Mg^2+^ deficiency [23,24]. Our research found that MGD significantly increased the starch content in leaves of *S. spontaneum* while having no significant effect on the starch content in the *S. officinarum* and *S. robustum* (Figure 1E). In *S. spontaneum*, the starch synthesis-related genes were significantly induced by MGD, while the amylolytic enzyme genes were inhibited under MGD conditions (Figure 4B). In contrast, these genes were slightly changed in *S. officinarum* and *S. robustum*. Therefore, the different expression patterns of starch biosynthesis and amylolytic enzyme genes likely cause the starch accumulation between *S. officinarum* and *S. robustum*.

Our data showed that MGD significantly reduced the photosynthesis efficiency in three *Saccharum* species (Figure 1C) and the carbohydrate export carrier in leaves was blocked (Figure 1E). In contrast, the life cycle essential processes such as ion homeostasis, sucrose biosynthetic process, vegetative to the reproductive phase transition of the meristem and cell death were significantly enriched in the modules negatively correlated with Mg concentration (Table 1). What is surprising is that in the category of ion homeostasis, *HKT1*, *MHX1* and *MGT10* were identified as MGD response genes (Figure 5I). Previous studies reported that HKT acted indirectly [17] while MHX directly [25] improved the ability of Mg absorption and *MGT10* plays a crucial role in Mg homeostasis in the chloroplast in *Arabidopsis* [26]. Our findings show that although MGD affects the growth and development of sugarcane, the stress response gene was up-regulated and prevented sugarcane from avoiding damage under MGD/HMT.

Lignin biosynthesis participates in plant growth, development and response to various biotic and abiotic stresses [27]. In *Arabidopsis*, lignin accumulation in roots prevents the excess efflux of metals from vascular cylinders rather than preventing uncontrolled influx [28]. In sugarcane, the lignin metabolic process was significantly enriched (Table 2, Appendix A) and the lignin content in roots (Figure 1F) was increased under MGD, suggesting that the lignin biosynthesis pathway genes up-regulated expression increased the degree of roots lignification under MGD. There might be at least two ways to improve the absorption efficiency of Mg^2+^ of sugarcane under MGD stress. The first is to increase the number of fibrous roots to enhance the absorption of water and nutrients by increasing the surface area of roots and the second is to increase the degree of root lignification and the lignin content, thereby preventing excess efflux Mg^2+^ from *Saccharum* plants.

The transcription factors, such as MYB, C2C2(Zn) DOF zinc finger, G2-like and WRKY, were up-regulated by MGD (Appendix A). It is reported that root hair growth by regulating auxin in *Arabidopsis thaliana* responds to environmental stress [29]. In *Brassica napus*, ectopic expression of Bn2R-MYBs could rescue the lesser root hair phenotype of the *Arabidopsis thaliana* and the Zinc finger protein 5 (ZFP5) was associated with the ethylene response to regulate the development of root hair under phosphorus and potassium deficiency environment [30]. In *Zea mays*, the homologous and known genes of *Arabidopsis thaliana* and *Oryza sativa* related to the architecture of roots such as AtERF109, AtERF73/HRE1, AtMYC2, OsMYC2 and AtWRKY6 probably play a role in the essential stages of roots growth and development [31]. Additionally, Ethylene and jasmonic acid could increase the lignin content by regulating and decreasing the cellulose content [32]. Combined with our data (Appendix A), we assumed that the TFs such as MYB and hormone signaling plays a minor role in regulating root hair development and lignin biosynthesis under MGD.

The genes belonging to Mg^2+^ concentration-associated modules lightgreen and magenta were significantly enriched in the lignin biosynthesis and metabolism processes. These two modules are highly correlated with lignin content (Table 2), suggesting that the disorder of lignin biosynthesis and metabolism caused by MGD may be a common sugarcane phenomenon. Moreover, we found that another functional category of the magenta module related to *S. officinarum* was enriched in the xylan biosynthetic processes (Figure 5B, Table 2). In *Arabidopsis*, a low level of xylan enhanced drought-tolerant ability and was sensitive to abscisic acid (ABA) [33]. Combined with previous lignin content detection results (Figure 1F), the lignin content in roots of *S. officinarum* did not increase significantly under MGD, probably because there are competed by xylan biosynthetic. Furthermore, the high content of xylan leads to the weakening of the stress response signal transmission, which may also be one of the reasons why *S. officinarum* does not possess high-stress resistance (Table 2).

Moreover, in roots, the NF-YA10 was identified as the hub gene in magenta which negatively associated with Mg^2+^ concentration (Figure 5K, Appendix A). Previous research indicated that NF-YA10 plays a role in stress adaptation [34,35,36] and regulates the length of the root apical meristem and lateral root density, suggesting that NF-YA10 probably plays a function in root development and increases the root hair to response to MGD stress in sugarcane. HB-7 was also identified as the hub gene that responds to magnesium deficiency (Figure 5K, Appendix A), it is reported that under alumina stress, AtHB7 and AtHB12 retro-regulated the resistance of alumina by affecting the accumulation of alumina in the cell wall, and HB-7 negatively regulated the growth of the roots in *Arabidopsis thaliana.* It seems possible that the decrease of main roots and increase in root hairs under MGD is due to the root development-related genes being down-regulated by MGD. The other hub genes were distributed in stress response (10) and transport (6) (Appendix A). Interestingly, CYP98A3 was identified as a hub gene in the roots of sugarcane, which involved lignin biosynthesis and flavonoid biosynthesis. In *Citrus sinensis*, MGD was upregulated in the genes involved in the lignin biosynthesis pathway, eventually increasing lignification in leaves [37]. To conclude, the TF probably has an essential role in response to MGD stress of sugarcane.

In the chloroplast biosynthesis pathway, CHLI2, is a subunit of Mg-chelatase, is involved in the assembly of the Mg-chelatase complex, and is required for chlorophyll biosynthesis [38]. In our study, CHLI2 was identified as a hub gene in the Mg^2+^ concentration positive module orangered4 module (Figure 5C), suggesting that CHLI2 plays the Mg-chelatase function in a MGS environment, while the expression of CHLI2 was down-regulated by MGD, likely leading to decreased biosynthesis in chloroplasts. Moreover, the STAY-GREEN-like gene is a hub gene that was identified in the violet module (Figure 5I, Appendix A). It is reported that the leaves display chlorosis under MGD stress which is caused by the expression of SGR in rice, which is retrogradely regulated by ROS production and promotes the degradation of chlorophyll. However, the chlorophyll degradation pathway of OsSGRL is not initiated by MGD [39]. This result suggested that SGRL is upregulated under MGD stress in sugarcane, which is probably one of the reasons for leaf chlorosis in sugarcane.

## 4. Materials and Methods

### 4.1. Plant Materials

Three founding *Saccharum* species, *S. spontaneum* (SES 208, 2*n* = 8x = 64, high resistance), *S. officinarum* (LA purple, 2*n* = 8x = 80 high sugar content and high sensitivity) and *S. robustum* (Molokai 6081, 2*n* = 64) [7] were planted on the campus of Fujian Agriculture and Forestry University (Fuzhou, China) for MGD treatment [40].

### 4.2. Magnesium Stress Treatment

According to previous study [21] and preliminary tests result, we determined the optimum Mg^2+^ concentration for growth of sugarcane is 0.2 mM. Therefore, this study utilized three Mg levels (0 mM Mg^2+^ treatment indicates MGD, 0.2 mM Mg^2+^ treatment indicates control condition and 1 mM Mg^2+^ treatment indicates HMT, respectively).

The sugarcane internodes were first removed to exclude residual nutritional influence from the stem and the node was impregnated with 1% carbendazim for three hours before treatment in hydroponic culture. The axillary buds shoot out and were moved to peat soil (pindstrup Substrate Seeding, pH 5.5, 0–10 mm) for planting. After two weeks, we picked plants with consistent growth conditions, and removed the stem nodes as much as possible. To reduce the effect of seedling residue on later experiments, we planted the *Saccharum* plants in the sand and pre-treated them for two months. During this time, tap water was applied for 10 days, then distilled water was used for 20 days to remove any nutritional residue from the sand and following this Hoagland nutrient solution with three different Mg^2+^ concentrations (0 mM, 0.2 mM, 1 mM) was provided for 30 days [41]. After 60 days of growth in Hoagland nutrition (including 60 days of hydroponic growth), the phenotype of each treated plant was determined, and the chloroplast content and photosynthetic efficiency were measured. Finally, sugarcane was sampled in different magnesium ion treatment groups and transcriptome sequencing samples were taken from each treatment of the six types of sugarcane.

### 4.3. Determination of Lignin Content

The acetyl bromide procedure was used to determine the relative lignin content of roots [42,43]. The samples (10 mg) were dissolved in 2 mL acetyl bromide reagent (25%, V/V) and heated to 70 °C for 30 min and 0.9 mL of 2 M NaOH, 3 mL of acetic acid and 0.1 mL of 7.5 M hydroxylamine were added in this order. Samples were centrifuged and 2 mL of the supernatant was removed. The absorbance of the supernatant at 280 nm was then measured by spectrophotometer [42,43].

### 4.4. Determination of Chlorophyll Content and Net Photosynthetic Efficiency

SPAD-502 Plus instrument (Konica Minolta Sensing Inc., Osaka, Japan) was used to measure the leaf chlorophyll content of the middle hypertrophic leaf (the first fully extended leaf) visible at the top. Photosynthetic measurements were performed on the sugarcane leaves of different treatments and the net photosynthetic efficiency and transpiration rate of each leaf was measured by a Li-6800 portable photosynthesis device (Li-COR, Lincoln, NE, USA), where the light intensity was set to 1000 μmol m^−2^ S^−1^ and CO_2_ R control 500 μmol^−1^. Three biological replicates were measured in each treatment and compared with the net photosynthetic efficiency and transpiration rate based on the obtained data.

### 4.5. Detection of Starch Content

Starch quantitative analysis was carried out according to the instructions of the Starch Determination Kit (Solarbio, Beijing, China, Cat#BC0700) on a spectrophotometer (Thermo Fisher Scientific OY Ratastie 2, FI-01620, Vantaa, Finland) using standard curves which were generated with different concentrations of glucose (0.01, 0.02, 0.03, 0.04, 0.05, 0.1 mg/mL). Each experiment had at least five biological replicates and similar results were obtained for each replicate. Error bars represent the SD of the mean [44].

### 4.6. RNA Extraction and Library Construction

Frozen samples were ground in liquid nitrogen. Total RNA was extracted using the TRIzol reagent. The RNA quality and quantity were determined using an Agilent 2100 Bioanalyzer (Agilent Technologies, Santa Clara, CA, USA). Both leaf and root samples have three independent biological replicates. We filtered the low-quality RNA and obtained samples to construct an RNA library for transcriptome sequencing. Each sample’s five microgram RNA was used to construct cDNA libraries, which were prepared using an Illumina^®^ TruSeq™ RNA Sample Preparation Kit (RS-122-2001(2), Illumina, San Diego, CA, USA) according to the manufacturer’s protocol. The RNA-seq libraries were pooled and sequenced with a Illumina Hiseq 2500 at the Novogene, Beijing, China. Clean data were aligned to reference gene models using HISAT [45] and the counts of genes were calculated using feature counts [46].

### 4.7. Gene Expression Analysis

We identified differential expressions in leaf and root tissues of three original *Saccharum* species by comparing gene expressions in samples of 0 mM vs. 0.2 mM and 0 mM vs. 1 mM genes. Genes with a *p*−value (FDR corrected) less than 0.05 and log2 fold change greater than 1 or less than −1 were considered up or down-regulated genes. Clustering of transcript expression patterns based on FPKM values, obtained as described above, was performed using the fuzzy c-means algorithm in the Mfuzz program (http://www.tm4.org/mev, accessed on 10 May 2021). Genes in each cluster were then classified according to GO terms and Fisher’s exact test was applied to test for enrichment of functional categories with Bonferroni’s correction (corrected *p* < 0.05).

### 4.8. Validation of FPKM Values for MGD Response Genes by qRT-PCR

RNA (≤1.0 μg) from each sample was reverse-transcribed to cDNA using a Reverse Transcriptase Kit (Takara, Beijing, China, Code, RR047A) in 20.0 mL reaction volume with 1.0 mL of random primers and 1.0 mL of mixed poly-dT primers (18–23 nt). This cDNA was diluted 1:19 in water for further qRT-PCR experiments. Specific primers were designed by Integrated DNA Technologies (https://sg.idtdna.com/pages, accessed on 10 December 2021) [47]. The TransStart Tip Green qRT-PCR SuperMix kit was used for qRT-PCR and the reaction cycle was: 95 °C for 30 s, 40 cycles of 95 °C for 5 s, 60 °C for 30 s and 95 °C for 10 s. The consistency of the melting curve demonstrated the reliability of qRT-PCR results. To normalize the expression levels, a constitutively expressed gene, eukaryotic elongation factor 1a (eEF-1a), was used as a reference gene. Each sample had four technical replicates. The relative expression levels for each different expression gene (DEG) in different magnesium treatments were calculated by the 2^−ΔΔCt^ method [48].

### 4.9. Gene Co-Expression Network Analysis

The co-expression networks were created using WGCNA (v1.66) package in R (Department of Human Genetics, University of California, Los Angeles, CA, USA). The genes with FPKM < 0.1 were discarded and a total of 20,572 genes were used as input to the assigned WGCNA network construction. The automatic network construction function block-wise was used to build modules. We set the soft power to 14 in all species DEGs WGNA analysis, 21 in leaves DEGs and 23 in roots. The min Module Size was set to 30 for all WGCNA analyses and the merge Cut Height value was 0.25. The initial clusters were merged on Eigengenes. An Eigengene value was calculated for each module and used to search for the association with flavor compounds. A soft threshold enabled WGCNA to preserve the continuous nature of the data set and eliminate the need to set an arbitrary correlation score cutoff. Here, the modules with a *p*-value ≤ 0.01with trait were thought to be trait-associated modules. Candidate hub genes in each module were chosen with the top 1/10-degree value genes. The co-expression network of each module was visualized using Cytoscape (v3.8.2, Institute for Systems Biology, Seattle, WA, USA).

## 5. Conclusions

This study combined physiological and RNA-seq analyses to identify potential mechanism responding to MGD of three *Saccharum* species. We found that the mechanism in response to MGD stress was significantly different among the three *Saccharum* species. GO analyses highlighted “stress response”, “chloroplast stroma”, “oxidation−reduction process” and “malate metabolic process” as key biological processes and metabolic pathways involved in MGD response in sugarcane. The accumulation of starch and lignin in leaves and roots caused by the genes involved in synthesis processes were up-regulated under MGD. Network analysis pinpointed several putative TFs from TCP, NAC, GATA, ERF, CO-like, G2-like and TALE families that could be essential regulators of the MGD response in sugarcane. Furthermore, several Mg transporters were identified that respond to MGD, the hub genes and these Mg transporters might be potential targets to engineer sugarcane plants with improved MGD tolerance. The involvement of these genes in response to Mg deficiency need to be confirmed by reverse genetic analysis, and the mechanisms underlying these responses to Mg deficiency deserve further investigation. This research provides a reference for the genetic modification of sugarcane varieties with high magnesium absorption efficiency in the future.

## Figures and Tables

**Figure 1 ijms-23-09681-f001:**
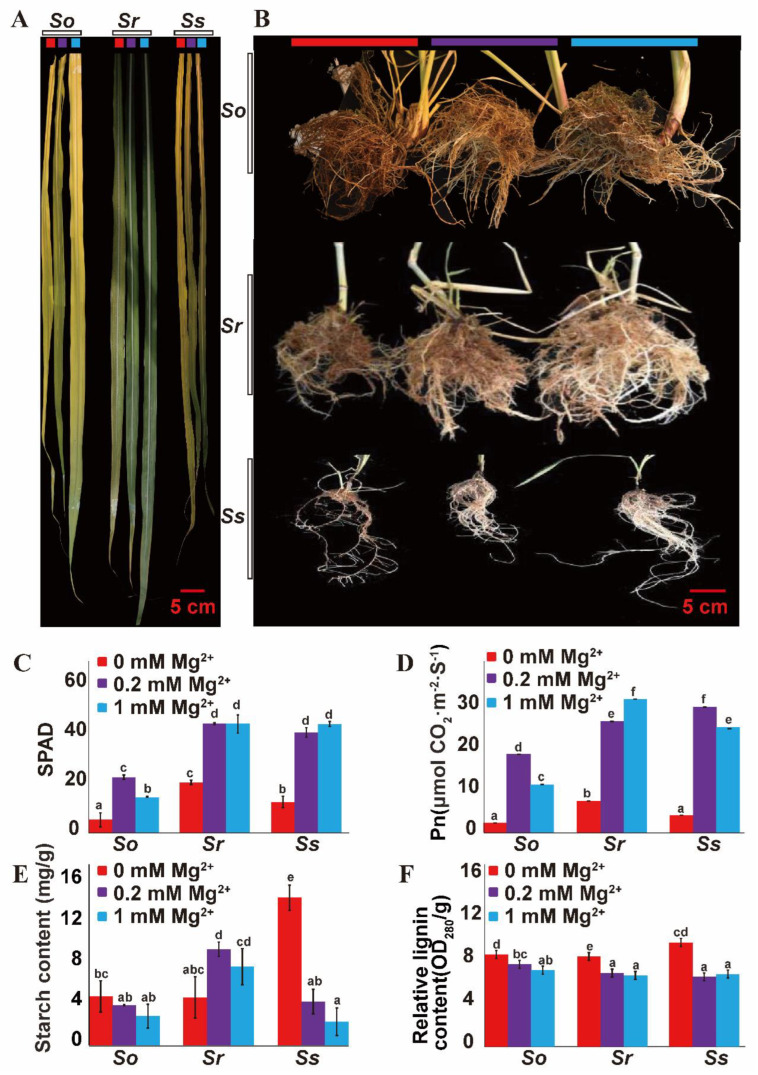
Phenotypes of three *Saccharum* species (**A**,**B**) and differences in net photosynthetic rates (**C**), SPAD values (**D**), starch content (**E**) and root lignin concentrations (**F**) under three different Mg^2+^ concentration treatments. Note: The red color indicates the 0 mM Mg^2+^ treatment, purple color indicates the 0.2 mM Mg^2+^ treatment, and blue color indicates the 1 mM Mg^2+^ treatment. *So* indicates *S. officinarum*; *Sr* indicates *S. robustum*; *Ss* indicates *S. spontaneum*; One−way ANOVA with Fisher’s test was used for statistical analysis (*p* < 0.05), and different letters expressed significant differences under three different Mg^2+^ concentration treatments.

**Figure 2 ijms-23-09681-f002:**
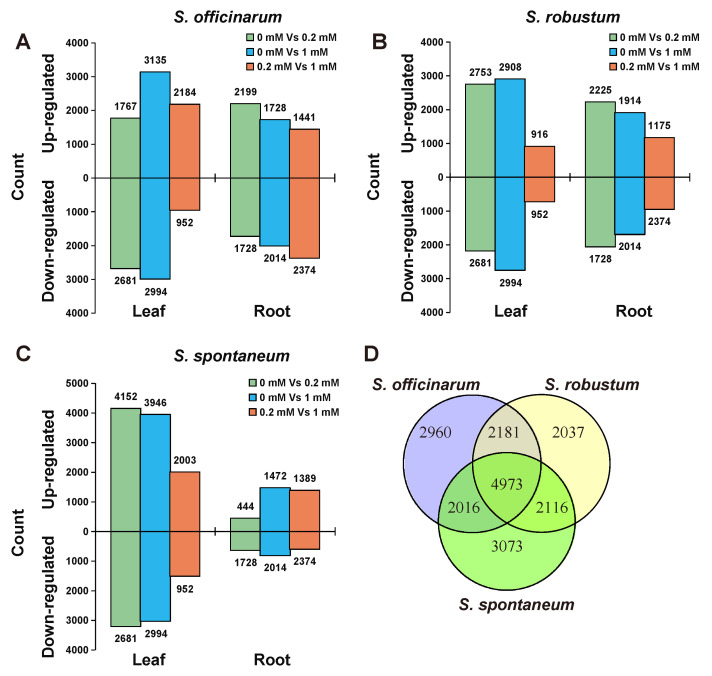
Differential gene expression analysis under three Mg concentration treatments of three *Saccharum* species. (**A**–**C**) Up- and down-regulated DEGs determined by the comparison of 0 mM vs. 0.2 mM and 1 mM in the leaf and root of three *Saccharum* species; (**D**) Venn diagrams showing unique and shared DEGs in three *Saccharum* species.

**Figure 3 ijms-23-09681-f003:**
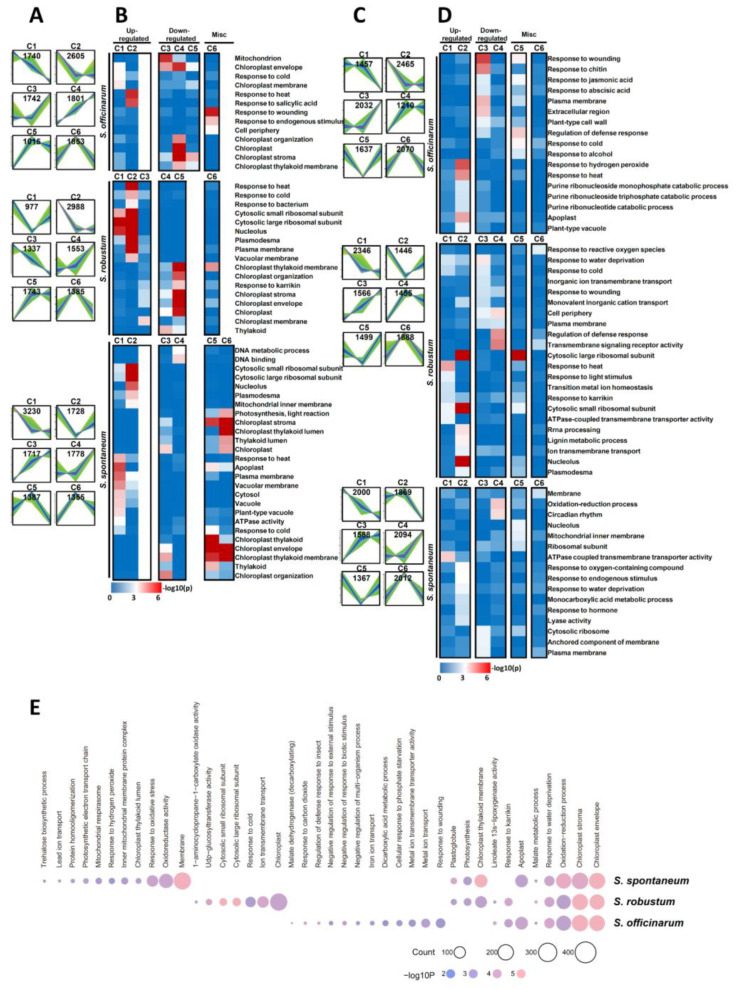
Dynamic progression of three *Saccharum* species transcriptome under different Mg concentration. (**A**,**C**) show the dynamic expression of three *Saccharum* species transcriptome in leaves and roots (C1−C6), respectively, by mufzz clustering. (**B**,**D**) show the functional enrichment among the clusters. (**E**) Functional enrichment among the DEGs of three *Saccharum* species. Note: C1−C6: Cluster 1−Cluster 6.

**Figure 4 ijms-23-09681-f004:**
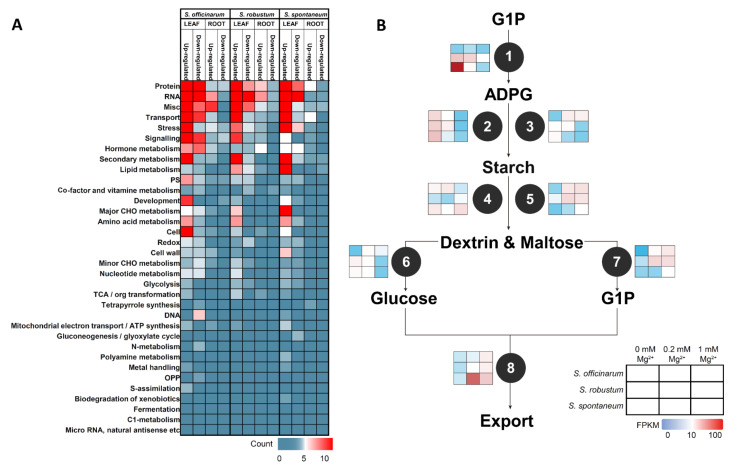
Mapman functional class enrichment for DEGs in leaves and roots of three *Saccharum* species (**A**) and schematic of the starch metabolism of sugarcane (**B**). The heat maps depict gene expression along with Mg concentration in leaves of three *Saccharum* species. 1, Glucose−1−phosphate adenylyltransferase; 2, Soluble starch synthase; 3, Starch branching enzyme; 4, Alpha−amylase; 5, Alpha−glucan water dikinase; 6, Glucose−1−phosphate adenylyltransferase; 7, Alpha−glucan phosphorylase; 8, Phosphate/phosphate translocator.Note: G1P: Glucose−1−phosphate; ADPG: Adenosine 5′−Diphosphoglucose.

**Figure 5 ijms-23-09681-f005:**
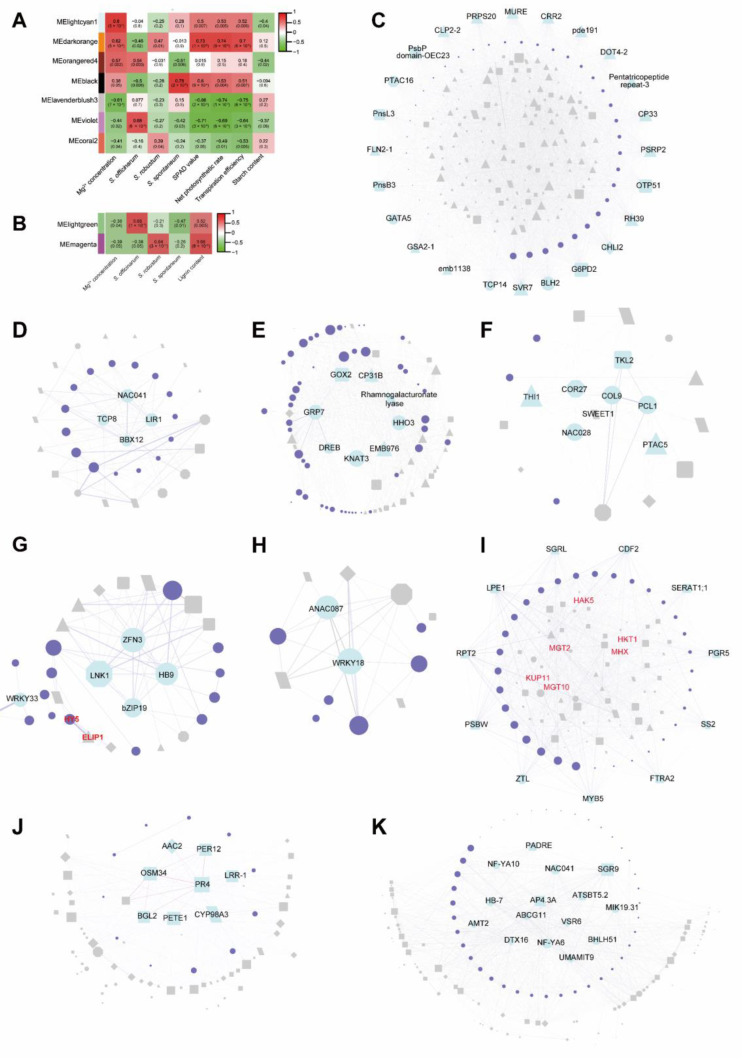
WGCNA analysis using the DEGs in leaves and roots of three *Saccharum* species. (**A**) heat map of module−trait correlation of Mg^2+^ concentration associated modules in roots; (**B**) heat map of module−trait correlation of Mg^2+^ concentration associated modules in leaves; (**C**–**I**) Network visualization of (**C**) orangered4, (**D**) lightcyan1, (**E**) black, (**F**) darkorange, (**G**) lavenderblush3, (**H**) coral2 and (**I**) violet modules, respectively, which associated with Mg^2+^ concentrations. (**J**,**K**) Network visualization of lightgreen and magenta modules which associated with Mg^2+^ concentrations.

**Table 1 ijms-23-09681-t001:** Gene ontology terms enriched for biological processes using the Mg^2+^ concentration associated modules DEGs in leaves.

Module	GO	Biological Process	Log10 (P)
lightcyan1 (R = 0.8, *p* value = 5 × 10^−7^)	GO:0070413	trehalose metabolism in response to stress	−5.7
	GO:0001666	response to hypoxia	−4.2
	GO:1905039	carboxylic acid transmembrane transport	−3.32
darkorange (R = 0.62, *p* value = 5 × 10^−4^)	GO:0014070	response to organic cyclic compound	−2.88
	GO:0007623	circadian rhythm	−2.46
	GO:0009737	response to abscisic acid	−2.37
orangered4 (R = 0.57, *p* value = 0.002)	GO:0009657	plastid organization	−21.44
	GO:0015979	photosynthesis	−16.04
	GO:0015994	chlorophyll metabolic process	−8.9
black (R = 0.38, *p* value = 0.05)	GO:0001666	response to hypoxia	−4.67
	GO:0000967	rRNA 5’ −end processing	−4.46
	GO:0009640	photomorphogenesis	−4.44
lavenderblush3 (R = −0.61, *p* value = 7 × 10^−4^)	GO:0010035	response to inorganic substance	−4.47
	GO:0044282	small molecule catabolic process	−4.14
	GO:0043648	dicarboxylic acid metabolic process	−3.78
violet (R = −0.44, *p* value = 0.02)	GO:0015849	organic acid transport	−5.64
	GO:0015979	photosynthesis	−5.45
	GO:0009639	response to red or far red light	−4.95
coral2 (R = −0.41, *p* value = 0.04)	GO:0042273	ribosomal large subunit biogenesis	−11.17
	GO:0002181	cytoplasmic translation	−8.91
	GO:0043648	dicarboxylic acid metabolic process	−4.9

**Table 2 ijms-23-09681-t002:** Gene ontology terms enriched for biological processes using the Mg concentration associated modules DEGs in roots.

Module	GO	Biological Process	Log10 (P)
lightgreen (R = −0.39, *p* value = 0.04)	GO:0009808	lignin metabolic process	−4.83
	GO:0045492	xylan biosynthetic process	−4.09
	GO:0044282	small molecule catabolic process	−3.98
magenta (R = −0.39, *p* value = 0.05)	GO:0009699	phenylpropanoid biosynthetic process	−6.28
	GO:0010035	response to inorganic substance	−5.95
	GO:0009308	amine metabolic process	−4.64

## Data Availability

All the expression data are available at our lab website (http://sugarcane.zhangjisenlab.cn/sgd/html/download.html, accessed on 5 June 2019).

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
