# Peer review of "Transcriptome Dynamics Underlying Magnesium Deficiency Stress in Three Founding Saccharum Species"

_ijms, 2022, doi:10.3390/ijms23179681_

Round 1

Reviewer 1 Report

This comparative study aimed to identify the effect of three different Mg2+ concentrations on the physiological and biochemical parameters of three Saccharum species. It was also performed a deeper transcriptome analysis under the same experimental conditions. The paper reveals interesting new insights that can be useful to understand better the relationship between the physiological and biochemical results observed and the genetic information behind them. The paper is generally straightforward and detailed, and the reading is fluent. Please, find below some of my considerations about some points of the paper I consider unclear.

·         Lines 81-84: There is no significant difference in the SPAD of Sorghum spontaneum between the treatment under 0.2 mM and the treatment under 0 mM of Mg2+. Considering the size of the bars, it could be considered suspicious. In addition, assuming that LMT would be 0 and 0.2 mM (please specify), it would be advisable to rephrase the statement.

·         Line 84: There is a repetition of the word “in.”

·         Fig. 1: It would be reasonable to be more specific. In particular, instead of “under different magnesium treatments,” it would be advisable to write “under three different magnesium concentration treatments.”

·         In the conclusion section would be helpful to put some information about future experiments and the scientific implications arising from this paper.

·         The difference between low magnesium concentration (LMT) treatments and high magnesium concentration treatment (HMT) does not seem clear. It seems that there is no specification about that in the introduction and in the material and methods. Specifically, which magnesium concentrations define the low and the high concentration treatment?

Is the low concentration treatments at 0 mM and 0.2 mM or only at 0.2 mM of Mg 2+

Reviewer 2 Report

The manuscript "Transcriptome dynamics underlying magnesium deficiency stress in three founding Saccharum species" reports the effect of Mg on gene expression in three species of Saccharum. The study is extensive and looked at different perspectives. However, there are flaws need to be taken care before publication. The term used next to the cultivars used confuses with the origin of the species, better use another term that expresses the specific source of the cultivars/genotypes used in the study. How the three levels of Mg treatment selected need to be clear. How 51 RNA-Seq libraries obtained is not clear.   I expected 54 based on the number of genotypes, Mg treatments and root and leaf tissues. What happened to three? The number of replication for the different experiments are different. Why? The result part is full of run-ons and need to be edited by a person English is native command. Some redundancy in statements observed (eg., L112-117). Some figures captions need to be clear (eg., abbreviations such as So, Sr, Ss were not clarified). It refers to the species but confusing to readers. The first sentence of your conclusion deviates from the results and discussion. 
